# Recent Advances in Nanotechnology with Nano-Phytochemicals: Molecular Mechanisms and Clinical Implications in Cancer Progression

**DOI:** 10.3390/ijms22073571

**Published:** 2021-03-30

**Authors:** Bonglee Kim, Ji-Eon Park, Eunji Im, Yongmin Cho, Jinjoo Lee, Hyo-Jung Lee, Deok-Yong Sim, Woon-Yi Park, Bum-Sang Shim, Sung-Hoon Kim

**Affiliations:** 1College of Korean Medicine, Kyung Hee University, Seoul 02447, Korea; bongleekim@khu.ac.kr (B.K.); wdnk77@naver.com (J.-E.P.); ji4137@naver.com (E.I.); ymcho@khu.ac.kr (Y.C.); leejinjoo1202@khu.ac.kr (J.L.); hyonice77@naver.com (H.-J.L.); simdy0821@naver.com (D.-Y.S.); wy1319@naver.com (W.-Y.P.); 2Korean Medicine-Based Drug Repositioning Cancer Research Center, College of Korean Medicine, Kyung Hee University, Seoul 02447, Korea

**Keywords:** nanoparticles, cancer progression, research milestones, clinical implications, nano-phytochemicals

## Abstract

Biocompatible nanoparticles (NPs) containing polymers, lipids (liposomes and micelles), dendrimers, ferritin, carbon nanotubes, quantum dots, ceramic, magnetic materials, and gold/silver have contributed to imaging diagnosis and targeted cancer therapy. However, only some NP drugs, including Doxil^®^ (liposome-encapsulated doxorubicin), Abraxane^®^ (albumin-bound paclitaxel), and Oncaspar^®^ (PEG-Asparaginase), have emerged on the pharmaceutical market to date. By contrast, several phytochemicals that were found to be effective in cultured cancer cells and animal studies have not shown significant efficacy in humans due to poor bioavailability and absorption, rapid clearance, resistance, and toxicity. Research to overcome these drawbacks by using phytochemical NPs remains in the early stages of clinical translation. Thus, in the current review, we discuss the progress in nanotechnology, research milestones, the molecular mechanisms of phytochemicals encapsulated in NPs, and clinical implications. Several challenges that must be overcome and future research perspectives are also described.

## 1. Introduction

Cancer progression is the result of tumor development and subsequent metastasis, with features including increases in the growth rate and invasiveness of tumor cells [1]. Though cancer therapies such as chemotherapy, radiotherapy, immunotherapy, and surgery have been used to suppress cancer progression for years, cancer treatment is still frequently unsuccessful due to poor solubility, low stability, limited biodistribution, metabolism, chemoresistance, and toxicity [2].

Hence, to overcome the poor bioavailability of anticancer agents, including natural compounds, nanobiotechnologies such as nanoparticles (NPs) have been employed for efficient drug delivery to mitigate poor solubility and stability, prevent degradation by proteases, improve drug distribution, and reduce drug resistance [3].

Accumulating evidence reveals that organic NPs (such as polymeric conjugates, polymeric NPs, lipid-based carriers (liposomes and micelles), dendrimers, and ferritin) and inorganic NPs (such as carbon nanotubes, quantum dots (QDs), ceramic NPs, magnetic NPs, and gold/silver NPs) are useful nanomaterials for achieving the enhanced permeability and retention (EPR) effect [4,5]. Furthermore, emerging evidence demonstrates that NPs that include a hydrophilic central core, a target-oriented biocompatible outer layer, and a middle hydrophobic core containing the target site can improve drug/gene delivery in cancer cells and tissues for ligand- or antigen-targeted therapy [6].

Though anticancer nanodrugs such as Doxil and Abraxane are on the pharmaceutical market, several nano-phytochemicals, defined as nanomaterials and phytochemicals, including curcumin [7] and EGCG [8], are attractive cancer therapy candidates, as experimental data suggest that they result in improved drug delivery and have low toxicity in several cancers. Thus, in this review, we discuss the recent progress of NP biotechnology, the molecular mechanisms of nano-phytochemicals, and their implications for the possible clinical application of potent anticancer nanodrugs on the basis of experimental studies in several cancers. Future research perspectives are also suggested.

## 2. Research Milestones in Nanotechnology and Anticancer Nanodrugs

Over the past decade, nanotechnology has greatly contributed to biomedical sciences, including oncology, with the development of efficient delivery systems. Colloidal gold particles were first synthesized as a typical hydrophobic colloid by Turkevich’s group in 1951 [9], and in 1965, Bangham’s group introduced liposomes as nanocarriers with composite structures made of phospholipids for transporting proteins and drugs [10]. Since then, polymeric NPs have included biodegradable and biocompatible polymers of synthetic (polylactide, polylactide–polyglycolide copolymers, polycaprolactones, and polyacrylates) or natural (alginate, albumin, or chitosan) origin. Wichterle et al. discovered hydrophilic gels in 1960, and Langer and Folkman were the first to introduce polymer systems appropriate for the controlled release of ionic molecules and macromolecules [11]. In 1978, Fritz Vögtle’s group became the first to synthesize dendrimers with structural stability [12]. Thereafter, Fraley et al. [13] reported the liposome-mediated delivery of DNA in 1980, and Gabizon et al. suggested that liposomal delivery improved the therapeutic index of encapsulated doxorubicin in 1982 [14]. In addition, QDs were synthesized by Ekimov’s group in 1982 [15], and in 1986, Matsumura and Maeda proposed the concept of the EPR effect, whereby nano-sized molecules accumulate more in tumor tissues than in normal tissues [16]. Among anticancer nanodrugs, the US FDA approved DOXIL (doxorubicin HCL liposome injection) for AIDS-related Kaposi’s sarcoma treatment in 1995 [17], DaunoXome (liposomal encapsulated daunorubicin) for HIV-related Kaposi’s sarcoma treatment in 1996 [18], Eligard (leuprolide acetate and polymer) for prostate cancer treatment in 2004 [19], Abraxane (albumin-bound paclitaxel injection) for metastatic pancreatic cancer treatment in 2005 [20], Oncaspar (Pegaspargase conjugated to mPEG) for acute lymphoblastic leukemia treatment in 2006 [21], Marqibo (vincristine sulfate liposome injection) for Philadelphia chromosome-negative lymphoblastic leukemia treatment in 2012 [22], and Onivyde (liposomal irinotecan) for pancreatic cancer treatment in 2015 [23]. Additionally, Myocet (non-PEGylated liposomal doxorubicin) was approved for metastatic breast cancer treatment in combination with cyclophosphamide in 2000 by the regulatory agencies of Europe and Canada [24]. Recently, anticancer nano-phytochemicals have been receiving increasing interest for their improved drug delivery and lower toxicity compared to synthetic anticancer agents. Among these compounds, gelatin/sugar-coated lycopene NPs were first characterized by Wegmann et al. in 2002 [25], and solid lipid NPs (SLNs) containing ferulic acid were characterized and developed as a sunscreen by Souto et al. in 2005 [26]. Additionally, Takahashi et al. prepared curcumin-loaded liposomes for encapsulation in 2006 [27], and in 2006, Hung et al. showed the NP potential of resveratrol coated with an emulsion-liposome blend, and no liver or kidney toxicity was detected [28]. Along the same line, systems with reported NP potential include gambogic acid-loaded micelles based on a chitosan derivative [29], ginsomes (ginsenoside-based NPs) for reinforcing the immune response of T and B lymphocyte in mice in 2009 [30], and EGCG encapsulated in PLA-PEG NPs [31] (Figure 1). Despite the advanced progress of nanotechnology over the past decade and a variety of potent nanodrugs or nano-phytochemicals, numerous questions and challenges remain for the future development of commercial anticancer nanodrugs. With this aim, this review focuses on a variety of NPs associated with phytochemicals, their drug delivery efficiency, and their anticancer efficacy. Then, future research directions are suggested.

## 3. Recent Advances in Nanotechnology Targeting Cancer Progression

A variety of NPs (1~100 nm) have been applied as efficient drug delivery systems, with advantages such as a high efficiency of drug loading and good bioavailability, drug delivery, and pharmacokinetics [32]. Generally, organic NPs include polymeric conjugates, polymeric nanoparticles, lipid-based carriers (liposomes and micelles), dendrimers, and ferritin, while inorganic NPs include carbon nanotubes, ceramic NPs, quantum dots, magnetic NPs, and gold/silver NPs [33] with specific nanostructure morphologies and sizes. Over the past decade, these NPs have been applied in the biomedical sciences for diagnosis and targeted drug delivery in cancer treatment (Figure 2).

### 3.1. Carbon-Based NPs

Carbon-based NPs are generally classified as single-walled carbon nanotubes (SWCNTs) and multiple-walled carbon nanotubes (MWCNTs) [34], which were first reported by Sumio Iijima [35,36] in 1991. These CNTs have attracted interest for their unique physicochemical properties that allow them to cross the cell membrane [37], as well as their highly versatile materials and enormous potential for biomedical applications [38]. However, CNTs are known to be cytotoxic in MC4L2 cells and mice. The antitumor effect of CNTs was confirmed in an animal model of breast cancer [39], and their increased cytotoxicity was attributed to the induction of oxidative stress in murine breast cancer [40]. Srivastava et al. [41] also demonstrated that MWCNTs induced oxidative stress and apoptosis in A549 cells. Similarly, Wang et al. [42] suggested that SWCNTs induced oxidative damage and apoptosis in PC-12 cells along with autophagy through the AKT–TSC2–mTOR pathway in non-small cell lung cancer cells (NSCLCs) [43]. Recently, Rh2 ginsenoside-hyaluronic acid-functionalized zinc oxide (Rh2HAZnO) was shown to have an enhanced antitumor effect compared to Rh2 alone in A549, HT29, and MCF7 cancer cells [44], and ZnO/carbon nanotubes were FDA-approved as safe in the body [45].

### 3.2. Ceramic NPs

Ceramic NPs (CNs) have emerged as drug delivery vehicles [46]. Silica nanoparticles (SNPs), discovered in 1992 by Kresge et al. [47], are known to have superior textual properties, such as a high surface area, narrow pore size distribution, large pore volume, and tunable pore diameter, for disease diagnosis and cancer imaging [48]. Recently, Yuan et al. [49] reported that SNPs encapsulating doxorubicin exerted antitumor effects through a burst release at an early stage and sustained release at a later stage. However, ceramic NPs have been reported to induce oxidative stress and inflammation in the lungs, liver, heart, and brain, leading to prethrombosis, genotoxicity, carcinogenicity, and teratogenicity, or brain toxicity [50]. Similarly, cerium oxide (CeO_2_) NPs induced apoptosis via ROS generation and p53-dependent mitochondrial signaling in HCT116 colorectal cancer cells [51]. Recently, *Cinnamomum cassia* extract was reported to protect against liver and kidney damage induced by nickel nanoparticles (Ni-NPs), which are used in applications such as ceramics and nanomedicine, in male Sprague Dawley rats [52]. Moreover, licorice isoliquiritigenin-encapsulated mesoporous silica nanoparticles (MSNs-ISL NPs) significantly suppressed receptor activator of nuclear factor-κB ligand (RANKL)-induced osteoclast generation, and the effect was greater than that achieved using ISL or MSNs alone [53].

### 3.3. Metal NPs

NPs containing metals such as gold, silver, copper, zinc, and palladium have unique physicochemical properties that are suitable for drug delivery and ligand targeting [54]. Among these complexes, gold NPs are regarded as bio-inert and non-cytotoxic and are used for medical applications such as MRI for imaging diagnosis and the photothermal treatment of cancer, in which NPs generate heat when exposed to near-infrared (NIR) laser light [55], with no acute cytotoxicity [56]. Indeed, Tsai et al. [57] reported that gold NPs inhibited proinflammatory cytokine production and TLR9 translocation via CpG oligodeoxynucleotides (CpG-ODNs), and Farooq et al. [58] demonstrated the anticancer and cytotoxic effects and intracellular localization of AuNPs in HeLa cells. Similarly, silver NPs have been used for biomedical applications such as infertility, antibacterial effects, skin damage, burns, and cancer treatment despite their harmful nanotoxicity [59,60]. Gurunathan et al. [61] reported that AgNPs, which are characterized by high yield, solubility, and stability, exerted dose-dependent cytotoxicity in MDA-MB-231 cells through the activation of lactate dehydrogenase (LDH), caspase-3, and reactive oxygen species (ROS) in breast cancer [62]. Similarly, B-AgNPs and F-AgNPs derived from *Bacillus tequilensis* and *Calocybe indica* (milky mushroom) extract induced apoptosis via the activation of p53, p-Erk1/2, and caspase-3 signaling and the downregulation of Bcl-2 in MDA-MB-231 breast cancer cells [62]. Interestingly, the release of Zn^2+^ from zinc peroxide nanoparticles (ZnO_2_ NPs) exerted an anticancer effect in a synergistic fashion with ROS production [63]. Ruenraroengsak et al. [64] demonstrated that mesoporous silica nanolayer (MSN)-ZnO-AuNSs reduced the viability of CAL51/CALDOX cells and MCF7/MCF-7-TX cells, while MSN-ZnO-AuNSs conjugated with Frizzled-7 (FZD-7) enhanced the toxicity by three-fold in resistant MCF-7TX cells. Further study on the efficacy and toxicity of metal NPs in animals and humans is required for their potential use in cancer diagnosis and therapy.

### 3.4. Quantum Dots

Quantum dots (QDs) are among the emerging engineering nanomaterials that have shown promise as a platform for cancer detection and diagnosis; examples include CdSe, ZnS, CdSe, ZnS, and CdS, which have unique optical and chemical properties [65]. Generally, colloidal QDs have been synthesized for diagnostic and therapeutic purposes in living systems through the processes of core development, shell growth, solubilization, and biological binding [66]. Lee et al. [67] reported that MNP-QD conjugates enhanced cellular uptake in HeLa cells without non-specific binding to the cell membrane, making them a promising new cell imaging technique. However, one drawback of QDs is their toxicity, which is caused by the release of factors that induce oxidative stress, including ROS, inflammatory cytokines, and metal ions [68]. Recently, curcumin quantum dots (CurQDs) were found to enhance the degradation of bacterial biofilms compared to curcumin alone [69], and folic acid [70] and chlorophyllin [71] have also emerged as promising QD NP candidates for imaging diagnosis.

### 3.5. Magnetic NPs

Magnetic NPs (MNPs) have been widely used for diverse applications, including magnetic biosensing (diagnostics), magnetic imaging, magnetic separation, drug and gene delivery, and hyperthermia therapy [72]. Interestingly, among MNPs, magnetic iron oxide NPs conjugated with integrin αvβ6 antibodies are known to have antitumor effects in oral squamous cell carcinoma [73]. Recently, galbanic acid-coated Fe_3_O_4_ MNPs were found to exert cytotoxic effects in PC3, LNCaP, and DU145 prostate cancer cells via the downregulation of AR, while galbanic acid was cytotoxic only in LNCaP cells [74].

### 3.6. Polymeric NPs

Polymeric NPs consisting of a reservoir system (nanocapsule) and matrix system (nanosphere) can incorporate hydrophilic/hydrophobic drug particles by coating inert materials in imaging, targeted drug delivery, and biomedical applications [75]. Known natural polymeric NPs include gelatin, chitosan, collagen, gum arabic, starch [76], dextran [77], alginate [78], and polylactic acid, while synthetic polymeric NPs in the form of dendrimers include N-(2-hydroxypropyl) methacrylamide (HPMA), polyglycolide or polyglycolic acid (PGA), PGA polyethylene glycol (PEG), polypropylenimine, polyamidoamine, and hydrogels [79,80]. Among the listed compounds, chitosan, a cationic polysaccharide, has been effectively used for the treatment of several cancers; its beneficial properties are biodegradability, mucoadhesiveness, enhanced absorption, and biocompatibility [81,82]. For instance, Amjadi et al. [83] suggested that DOX@BET-loaded PGNPs—betanin (BET) and doxorubicin (DOX) encapsulated by gelatin nanoparticles (GNPs)—showed better anticancer activity in MCF-7 cells compared to DOX or BET alone. Similarly, pretreatment with collagozomes followed by paclitaxel micelles facilitated drug penetration into pancreatic ductal adenocarcinoma, resulting in better antitumor activity [84], while gum arabic-encapsulated gold NPs (GA-AuNPs) combined with the application of lasers induced cell death in lung tumor tissues via the inhibition of inflammation, angiogenesis, and lipid peroxidation [85]. Furthermore, Babu et al. [86] suggested that the combinatorial delivery of cisplatin and p62siRNA/β5 plasmid DNA-mediated chitosan-coated polylactic acid NPs enhanced the antitumor effect of cisplatin in resistant ovarian cancer cells. Additionally, Strong et al. [87] reported that polyhydrogels with silica–gold nanoshells loaded with either doxorubicin or a DNA duplex enhanced drug release by 2–5-fold after exposure to NIR light in CT 26-WT colon cancer cells.

### 3.7. Lipid-Based NPs

Lipid-based NPs with low toxicity include liposomes, micelles, solid lipid nanoparticles (SLNs), and nanostructured lipid carriers (NLCs) [88,89]. However, a major drawback of SLNs is their low drug loading capacity due to the release of the loaded drug solution [90]. NLCs have a less defined lattice defect with an incomplete crystal or amorphous structure. Consequently, drug loading occurs at the defect site, leading to greater accumulation and less excretion of the drug, since the compartment is surrounded by solid lipids [91,92]. Liposomes have a bilayer structure that is mainly composed of phospholipids, and they can encapsulate both hydrophobic and hydrophilic drugs because of the amphiphilic properties of phospholipids. Furthermore, liposomes can improve stability and efficiency by preventing the drug from being metabolized [93,94]. Notably, lipid NPs with aqueous cores can be used as carriers of hydrophilic drugs, mainly for oral delivery, while polymeric NPs, including carbon nanotubes, dendrimers, and quantum dots, can encapsulate water-insoluble drugs in their hydrophobic cores for injectable drug delivery systems [95]. Wang et al. [96] suggested that myricetin nanoliposomes (MYR-NLs) enhanced apoptosis in DBTRG-05MG glioblastoma multiforme (GBM) cells by reducing glycolysis. Similarly, Maroufi et al. [97] reported that myricetin-loaded nanostructured lipid carriers (NLCs) enhanced the anticancer effect of DXT via the downregulation of Mcl-1, survivin, and cyclin B1 and the upregulation of Bid and Bax in MDA-MB231 breast cancer cells.

### 3.8. Dendrimers

Dendrimers are manufactured using several macromolecules, including polypropyleneimine (PPI), polyamidoamine (PAMAM), poly-L-lycine (PLL), melamine, triazine, and poly(ethylene glycol) (PEG) [98]. Dendrimers are suitable for loading hydrophilic and hydrophobic drugs and are characterized by branches, distinct molecular weights, and a globular assembly with a meticulous surface, and they have been studied since Dr. Donald Tomalia first published his work on poly(amidoamine) (PAMAM) dendrimers in 1985 [99]. Indeed, Dickwalkar et al. [100] reported that polyamidoamine dendrimers (PAMAMG_4.0_-NH_2_) conjugated with the omega-3 fatty acid docosahexaenoic acid (DHA) and paclitaxel (PTX) enhanced the anticancer activity of PTX compared to PTX or PAX (PAMAMG_4.0_ -NH_2_ -PTX) alone in upper gastrointestinal cancer cells. Recently, Mignani et al. [101] suggested that several anticancer phytochemicals, including epirubicin (4′-epi-isomer of DOX), methotrexate, daidzein, genistein, 7-ethyl-10-hydroxycamtothecin (SN-38), colchicine, 7-butyl-10-aminocamptothecin, lamellarin D, digoxin, biotin, a tubulysin D analog, and 10-hydroxycamptothecin, could be conjugated only with dendrimers for enhanced anticancer activity. Accumulating evidence also suggests that dendrimers combined with phytochemicals can serve as drug and gene carriers that impart good solubility and bioavailability to hydrophobic drugs in cancer therapy [101,102]. Aas et al. [103] reported that dendrosomal farnesiferol C (DFC) significantly enhanced the anti-proliferative effect in a time- and dose-dependent manner compared to FC alone in AGS cells.

### 3.9. The Enhanced Permeability and Retention Effect

The enhanced permeability and retention (EPR) effect first coined by Matsumura and Maeda [16] is defined the phenomenon of macromolecules or high molecular weight drug and nanomedicine accumulation inside solid tumor models compared to healthy tissue counterparts [4]. EPR effect is usually induced by a leaky tumor vasculature by the accelerated angiogenesis and impaired lymphatic drainage by the disorganized growth of tumors [104,105]. In details, NPs with appropriate sizes can evade the tumor capillaries and be retained in the tumor tissues for days due to the lack of lymphatic drainage. Additionally, particles with high positive charges can bind non-specifically to the negatively charged luminal surface due to the presence of sulfate and carboxylate sugar moieties [106]. It is noteworthy that nitric oxide (NO), prostaglandins, and bradykinin, which act as vasodilators, can enhance the EPR effect in tumors by increasing their vascular permeability [107]. Furthermore, some nanomedicine formulations are effective in the treatment of multidrug resistance (MDR) [108]. In contrast, Jain et al. [109] claimed that elevated interstitial fluid pressure and heterogeneous blood supply limit macromolecular delivery to tumors. Furthermore, a key challenge is the promotion of the EPR effect in patients with EPR-insensitive tumor phenotypes since the EPR-insensitive phenotype is known to have smaller endothelial fenestrations, heterogeneously high or low pericyte coverage, more developed and branched vasculatures, a relatively dense ECR, and more developed immune profiles compared to EPR sensitive phenotypes [105]. Thus, the role of EPR is still in question in terms of clinical translation and different human tumor types due to their heterogeneity [105,110]. Among the three major targeted drug delivery methods, namely, passive targeting, active targeting, and physical targeting; passive targeting acts via the EPR effect, by which tumor cells preferentially absorb NPs [111]. In active targeting, NPs that are functionalized with ligands such as proteins, antibodies, and peptides effectively interact with overexpressed receptors at the target site [112]. Physical targeting utilizes external sources or fields, such as radiation, ultrasound, photothermal and magnetic hyperthermia therapies, to guide NPs to the target site and control drug release through changes in pH and/or temperature especially in EPR-insensitive tumor phenotypes [103,106]. Thus, Theek et al. [113] revealed that increased accumulation of liposomes was shown in two EPR-insensitive phenotypes such as highly stromal BxPC-3 pancreatic carcinoma xenograft and highly cellular A431 epidermal xenograft after ultrasound irradiation compared to untreated controls.

### 3.10. The Reticuloendothelial System Barrier in Nanoparticle Drug Delivery

One of the clinical translation issues with NP drugs has been the reticuloendothelial system (RES) barrier because hepatocytes and Kupffer cells in the RES in the liver usually take up NPs bound to serum proteins, depending on their size and surface properties [103,114]. Thus, for the purpose of clinical translation, a good strategy to block or deplete macrophages is to boost the efficiency of nanoparticle drugs. Recently, Tang et al. suggested that d-self-peptide-labeled liposomes (DSLs) could reduce interactions between phagocytes and NPs by forming a long-lasting mask [115] since the modification of NPs with polyethylene glycol (PEG) reduced their clearance by macrophages and their internalization [116]. Similarly, multicore iron oxide NPs coated with a poly(4-vinylpyridine) polyethylene glycol copolymer have low RES retention and high urinary excretion in the kidneys [117].

## 4. Molecular Mechanisms of Nano-Phytochemicals in Cancer Progression

Recently, phytochemicals, including functional food and anticancer supplements, have emerged as effective anticancer agents for cancer therapy with fewer side effects than conventional treatments [118]. Among phytochemicals, curcumin, epigallocatechin gallate (EGCG), ginsenosides, lycopene, and resveratrol are more attractive than others. In particular, curcumin, the Indian spice turmeric from *Curcuma longa*, has been shown to exert an anticancer effect in several cancers, including breast, colon, and pancreas, brain, and liver cancers [119]. Despite its beneficial effects, its therapeutic efficacy is limited due to poor bioavailability, poor absorption, rapid metabolism, and rapid systemic elimination [120,121,122]. Thus, to overcome the shortcomings of natural compounds such as curcumin, several types of NPs have been applied with phytochemicals in cancer therapy, as shown in Table 1.

### 4.1. Anacardic Acid

Anacardic acid (AA) from *Anacardium occidentale* has antitumor effects in several cancers. AA inhibited proliferation, invasion, and migration and induced G0/G1-phase arrest and apoptosis in MDA-MB-231 cells via the inhibition of Hsp90-dependent endoplasmic reticulum stress (ERS)-related molecules, such as GRP78, Hsp70, CDK-4, MMP-9, Bcl-2, and Mcl-1 [123]. To increase the efficacy of this molecule, nanotechnology with AA has been adopted [124]. Kushwah et al. [125] reported that docetaxel (DTX)-loaded bovine serum albumin (BSA) covalently conjugated with AA and gemcitabine (GEM) nanoparticles (AA-GEM-BSA NPs) significantly improved the cellular uptake, apoptosis, and pharmacokinetic profile in MCF-7 and MDA-MB-231 breast cancer cells compared to DTX and GEM alone. Specifically, AA-GEM-BSA NPs were associated with significantly sustained release, enhanced stability against enzymatic degradation, delayed DTX release, and higher levels of the apoptosis index in MCF-7 and MDA-MB-231 cells as compared to a combination of GEM and DTX. Thus, the combination of AA and DTX or GEM encapsulated by BSA protein is a promising candidate for breast cancer treatment.

### 4.2. Betulinic Acid

Though betulinic acid (BA), a pentacyclic triterpenoid derived from birch tree, has a broad spectrum of biological and medicinal properties, it has some drawbacks, such as poor aqueous solubility and a short half-life in vivo [126]. Thus, several types of nanoscale delivery systems, such as polymeric NPs, magnetic NPs, liposomes, polymeric conjugates, nanoemulsions, cyclodextrin complexes, and carbon nanotubes, have been developed for the efficient delivery of BA [127]. Saneja et al. revealed that GEM -BA NPs significantly enhanced cytotoxicity and ROS generation in pancreatic cancer Panc-1 cells compared to GEM NPs or GEM and BA or GEM alone [128]. Farcas et al. [129] indicated that BA-loaded magnetoliposomes, as a potent anticancer agent, enhanced the antitumor effect in MDA-MB-231 breast cancer cells without significant cytotoxicity in normal breast epithelial MCF 10A cells. Similarly, Kumar et al. [130] reported that poly(lactic- co-glycolic acid)-loaded NPs of BA (PLGA-loaded NPs of BA) significantly decreased the expression of i-NOS, Bcl-2, and Bcl-xl and increased the expression of BAD, BAX, and caspase-9/3 compared to the untreated control in a diethylnitrosamine (DEN)-induced hepatocellular carcinoma (HCC) rat model. Overall, polymeric or lipid NPs with BA are promising candidates for treating breast, pancreas, and liver cancers and have outperformed BA controls, indicating that further study on BA-NPs is required in humans to determine their pharmacodynamic and pharmacokinetic profiles.

### 4.3. Curcumin

It is well known that curcumin (diferuloylmethane:CU)—a hydrophobic polyphenol—modulates enzymes, transcription factors, kinases, inflammatory cytokines, growth factors, and proapoptotic and antiapoptotic proteins in several cancers [119]. Though CU has been shown to be a potent anticancer compound in numerous cancers in vitro and in vivo, several clinical trials have found that its efficacy is limited in humans because of characteristics that reduce its bioavailability, such as low water solubility, poor absorption, a high rate of metabolism, the inactivity of metabolic products, and/or rapid elimination and clearance from the body [120,131]. As one of the approaches to overcome the disadvantages of CU, NPs that include liposomal encapsulation and emulsions have been employed for many years [132]. Li et al. [133] indicated that liposomal CU enhanced cytotoxicity and apoptosis in LoVo and Colo205 colorectal cancer cells and exerted significant (*p* < 0.05) synergistic effects with oxaliplatin in a colorectal xenograft model via the inhibition of CD31, VEGF, and IL-8 and the activation of PARP cleavage. Similarly, Pandelidou et al. [134] demonstrated that egg phosphatidylcholine (EPC) liposomes conjugated with CU showed significantly better cytotoxicity compared to the CU control in HCT116 and HCT15 colorectal cancer cells. Notably, Arya et al. [135] demonstrated that CU-loaded chitosan/polyethylene glycol (PEG)-blended PLGA NPs exhibited significant cytotoxic, anti-invasive, and apoptotic effects compared to the CU control in Panc-1 and Mia Paca-2 pancreatic cancer cells via the activation of BAX, caspase-3, and PAPR cleavage and the inhibition of BCL-2. Moreover, Khan et al. [136] reported that CU-loaded chitosan nanoparticles (CLCsNPs) produced better antitumor effects in cervical cancers compared to chitosan NPs (CsNPs) alone. Furthermore, CU-resveratrol-gelucire 50/13 (CRG) SLNs had better cytotoxicity in HCT116 cells compared to CU-resveratrol-gelucire 50/13-HPβCD (CRG-CD) [137] due to their greater bioavailability, and the adverse effects were limited [131]. Overall, CU NPs have shown significant anticancer activity in several cancers and are promising as potential combinatorial agents with classical anticancer drugs.

### 4.4. EGCG

Accumulating evidence has confirmed that epigallocatechin gallate (EGCG), a flavone-3-ol polyphenol from green tea, is mainly absorbed in the intestine [138] and inhibits NF-κB, epithelial–mesenchymal transition (EMT), and cellular invasion in several cancers via its interaction with DNA methyltransferases (DNMTs), histone deacetylases (HDACs), Pin1, TGFR-II, MMP-2, and MMP-9. In an aim to improve the insolubility, instability, and low tissue distribution of EGCG, Chu et al. [139] investigated polymeric CU/EGCG-loaded NPs in an orthotopic prostate tumor model. The NPs exerted stronger antitumor effects compared to the CU/EGCG combination control by targeting CD34 and P-selectin since hyaluronic acid targets CD44 and fucoidan targets P-selection in tumor vasculature. Zhang et al. [140] demonstrated that, compared to EGCG alone, polymeric EGCG-loaded PLGA NPs had enhanced anticancer activity in A549 and H1299 lung cancer cells. The anticancer effects occurred via the inhibition of NF-κB and its related proteins, such as Bcl-2, Bcl-xL, COX-2, TNF-α, cyclin D1, c-Myc, TWIST1, and MMP-2. Similarly, Velavan et al. [141] demonstrated that BSA-encapsulated magnetite nanoparticles (MNPs) loaded with EGCG (nano-EGCG) showed better antitumor efficacy in A549 lung cancer cells compared to EGCG alone via increased ROS/RNS modulation of Nrf2/keap1 signaling and the loss of the mitochondrial membrane potential, leading to apoptosis. Additionally, Radhakrishnan et al. [142] indicated that bombesin-conjugated EGCG-loaded SLNs resulted in increased cytotoxicity in MDA-MB-231 breast cancer cells and reduced the tumor size of B16F10 cells compared to EGCG or the bombesin-conjugated EGCG (EB-SLN) group. Similarly, Hajipour et al. [143] reported that arginyl-glycyl-aspartic acid (RGD)-containing ECGC-loaded nanostructured lipid carriers (EGCG-loaded NLC-RGD) enhanced the cytotoxic and apoptotic effects of doxorubicin (DOX) in MDA-MB-231 breast cancer cells compared to EGCG-loaded NLCs. Notably, Chavva et al. [144] demonstrated that EGCG–gold nanoparticles (E-GNPs) showed better cytotoxicity and cellular uptake in PC-3 and MDA-MB-231 cells compared to free EGCG or citrate GNPs via the inhibition of NF-κB, BCL-2, and BCL-xL and the activation of Bax and cleaved caspase-7/3. Overall, though some nanoparticles with EGCG have been found to be effective in vitro and in vivo by compensating for the weaknesses of EGCG, further study with EGCG-NPs is required to establish the pharmacodynamic and pharmacokinetic profiles for future clinical application.

### 4.5. Ferulic Acid

Ferulic acid (FA), which is commonly found in vegetables, sweet corn, bamboo shoots, and rice grain [145], is known to have cytotoxic and apoptotic effects via the inhibition of PI3K/AKT signaling in pancreatic and cervical cancers [146,147]. Recently, to improve the EPR effect of FA, Cui et al. [148] studied the antitumor effects of FA-modified selenium nanoparticles (FA-Se NPs) in HepG2 cells and reported increased ROS production, MMP disruption, and caspase-9/3 activation compared to FA alone, while ZnO nanoparticles (ZnONPs)-FA (FAC), compared to ZnONPs or FA alone, suppressed hepatocellular cancer progression via ROS production; the inhibition of MMP-2 Bcl-2, and Bcl-xL; and the activation of Bax, Bad, cleaved caspase-3, and cleaved PARP [149]. Furthermore, Thakkar et al. [150] reported that the chemopreventive effect of FA and aspirin (ASP) encapsulated by chitosan-coated solid lipid nanoparticles (c-SLNs) occurred via the inhibition of PCNA and Ki67 and the activation of p-RB, p21, and p-ERK1/2 in MIA, PaCa-2, and Panc-1 pancreatic cancer cells. Further study is required in humans to determine the PD and PK profiles of FA-NPs, both alone and in combination with classical anticancer drugs, for future clinical trials.

### 4.6. Gambogic Acid

Gambogic acid (GA), a component of the dry resin (gamboge) from the *Garcinia hanburyi* tree, is known to have antitumor effects in prostate [151], lung [152], stomach [152], and liver [153] cancers. GA induces apoptosis in cancer cells by increasing ROS production and inhibiting the NF-κB, MAPK/ERK, and PI3K/AKT signaling pathways. Studying polymeric NPs with GA, Xu et al. showed that the apoptotic effect of L-methionine poly(ester amide) (Met-PEA-PEG) via high intracellular ROS production in PC-3 cells was better than that of GA alone [154]. Furthermore, Wang et al. [155] demonstrated the improved apoptotic effect of hyaluronic acid-grafted polyethylenimine-poly(D,L-lactide-co-glycolide) (HA-PEI-PLGA) NPs via the activation of caspase-3/8 and the inhibition of survivin and Bcl-2 in triple-negative breast cancer (TNBC) compared to GA alone. Similarly, He et al. [156] reported that GA-loaded folate-conjugated Arg-based poly(ester urea urethane)s NPs (GA-loaded FA-Arg-PEUU NPs) induced more powerful apoptotic and anti-metastatic effects via the inhibition of MMP-2/9 and mitochondrial membrane potential and the activation of DNA fragmentation compared to GA alone in HeLa and A549 cells. Notably, Fang et al. [157] showed that magnetic NPs containing Fe_3_O_4_ as a carrier for GA (GA MNPs-Fe_3_O_4_) had a stronger apoptotic effect in LoVo colon cancer cells compared to GA alone via the activation of caspase-9/3 and the suppression of p-PI3K, p-AKT, and p-BAD. Overall, polymeric or metal NPs with GA can enhance drug delivery and thereby increase anticancer activity compared to GA alone and should be studied in future clinical trials.

### 4.7. Ginsenosides

Ginsenosides or panaxosides are a class of steroid glycosides and triterpene saponins derived from *Panax ginseng*. The major protopanaxadiols include ginsenoside Rb1, Rb2, Rg3, Rh2, and Rh3, while the major protopanaxatriols are ginsenoside Rg1, Rg2, and Rh1 [158,159]. To increase their low oral bioavailability and achieve efficient drug release, several NPs, including liposomes, emulsions, and micelles, have been conjugated with ginsenosides for cancer therapy [160]. Ren et al. [161] reported that Fe_3_O_4_ NPs with ginsenoside Rg3 (NpRg3) significantly inhibited HCC development and metastasis via the remodeling of unbalanced gut microbiota and metabolism compared to Rg3 or Fe_3_O_4_ alone. Similarly, Qiu et al. [162] reported that Rg3 co-loaded with poly(ethylene glycol)-block-poly(L-glutamic acid-co-L-phenylalanine) (mPEG-b-P(Glu-co-Phe)) NPs (Rg3-NPs) significantly reduced proliferating cell nuclear antigen (PCNA) and increased caspase-7/9/3 compared to Rg3 alone in a colon cancer xenograft mouse model. Kim et al. [44] revealed that Rh2 hyaluronic acid-functionalized zinc oxide (Rh2HAZnO) enhanced p53, pp38, Bax, PARP, and ROS production and reduced the expression of Bcl-2 compared to Rh2 alone in A549 lung cancer, HT29 colon cancer, and MCF7 breast cancer cells. Additionally, Dong et al. [163] demonstrated that folic acid (FA)-modified BSA and Rg5 NPs (FA-Rg5-BSA NPs) significantly enhanced cellular uptake and apoptotic cell death compared to the Rg5 or Rg5-BSA NP group in MCF-7 cells and a xenograft mouse model. Overall, several ginsenosides encapsulated by NPs have been found to exert anticancer effects by increasing the drug delivery efficiency. However, further study in humans is recommended to analyze the pharmacodynamic and pharmacokinetic profiles for clinical applications and the scale-up of ginsenosides, either alone or in combination with classical anticancer drugs.

### 4.8. Kaempferol

Kaempferol (3,4′,5,7-tetrahydroxyflavone; KPF), a natural flavonol, is present in many edible plants, such as tea, kale, beans, leek, tomato, strawberries, broccoli, cabbage, and grapes, and common medicinal plants (*Aloe vera, Ginkgo biloba, Rosmarinus officinalis, Crocus sativus, Hypericum perforatum*) [164]. Though KPF is known to have antioxidant, anti-inflammatory, estrogenic, anxiolytic, analgesic, antimicrobial, cardioprotective, neuroprotective, antidiabetic, and anticancer activities [164,165], it has some shortcomings, such as poor drug delivery due to low water solubility and bioavailability in clinical application [95]. To improve the bioavailability of KPF, it has been used in several types of NPs in cancer cells. Luo et el. [166] comparatively evaluated the efficacy of five different types of NPs with the incorporation of KPF, such as poly(ethylene oxide)-poly(propylene oxide)-poly(ethylene oxide) (PEO-PPO-PEO poly(DL-lactic acid-co-glycolic acid) (PLGA), PLGA-polyethylenimine (PEI), glycol chitosan, and poly(amidoamine) (PAMAM) dendrimer in ovarian cancer cells. In A2780/CP70 and OVCAR-3 ovarian cancer cells, PEO-PPO-PEO and PLGA NPs reduced cell viability, whereas PLGA-PEI, glycol chitosan, and PAMAM dendrimer NPs did not show significant efficacy, implying that PLGA NPs may be a promising candidate for cancer treatment. Additionally, Govindaraju et al. reported that KPF-conjugated gold nanoclusters (K-AuNCs) had stronger cytotoxicity in A549 cells compared to KPF alone, and they did not damage HK-2 normal kidney cells [167]. Notably, Chuang et al. [168] suggested that gelatin nanoparticles (GNPs) with KPF encapsulation (GNP-KA) inhibited the viability and migration of human umbilical vein endothelial cells (HUVECs) and suppressed vessel formation in mice cornea compared to KPF alone via the inhibition of MMP-2, MMP-9, and VEGF. Interestingly, Colombo et al. [169] reported that a KPF-loaded mucoadhesive nanoemulsion (KPF-MNE) more effectively permeated the mucosa, and the retention of KPF in the nasal mucosa was significantly improved, making it an effective therapeutic candidate for glioma treatment. Overall, NPs with materials such as polymers, metals, lipids, and gelatin encapsulated with KPF enhanced the antitumor efficacy of KPF without harming normal cells, implying that polymeric, lipid, metal, and gelatin NPs with KPF are potent antitumor candidates for cancer therapy.

### 4.9. Lycopene

Lycopene (LYC), a member of the carotenoid family that naturally occurs in tomatoes and guava watermelon, is known to have anticancer activity in prostate, colorectal, and gastric cancers [170]. To enhance its efficacy and bioavailability, several types of NPs have been used with lycopene in cancer therapy [171]. Vasconcelos et al. [172] demonstrated that poly(ε-caprolactone) lipid-core NPs containing lycopene-rich extract from red guava (nano-LEG) were more effective and safe in MCF-7 breast cancer cells compared to the LYC control via the inhibition of NF-kB activation and ROS production. Similarly, Jain et al. [173] reported that lycopene-loaded solid lipid NPs (LYC-SLNs) exerted a synergistic effect with methotrexate in MCF-7 breast cancer cells, and cellular uptake of LYC-SLNs was higher compared to LYC alone. Furthermore, Jain et al. [173] demonstrated that LYC-loaded whey protein NPs (LYC-WPI-NPs) enhanced the bioavailability and prophylactic effect of LYC in MCF-7 breast cancer cells compared to LYC alone. Similarly, the results reported by Huang et al. [174] suggest that a nanoemulsion of LYC and gold nanoparticles (AN) has potential as a potent therapeutic agent against colon cancer. The AN–LYC nanoemulsion significantly reduced the expression of procaspases-8, -3, and -9, PARP-1, Bcl-2, Akt, NF-κB, and pro-MMP-2/9 and enhanced the expression of Bax and E-cadherin. Overall, lycopene encapsulated with lipid, metal, and polymeric NPs appear to be more potent antitumor agents than lycopene alone in prostate, ovarian, lung, and breast cancers.

### 4.10. Resveratrol

Resveratrol (RES), a non-flavonoid polyphenol and a phytoestrogen, is known to have antioxidant, anti-inflammatory, cardioprotective, anticancer, and anti-multidrug resistance (MDR) effects [175,176]. However, since the poor bioavailability of resveratrol in humans has been a critical issue, recent studies have focused on its delivery system, formulation, and interaction with other compounds [177]. Senthil Kumar et al. [178] demonstrated that chitosan-coated-transresveratrol (RSV) and ferulic acid (FER)-loaded solid lipid nanoparticles (SLNs) conjugated with folic acid (FA) (C-RSV-FER-FA-SLNs) exhibited enhanced cytotoxic activity and G0/G1 stage arrest compared to free RSV-FER or RSV-FER-SLNs in HT-29 colon cancer cells. Furthermore, Park et al. [179] reported that resveratrol-capped gold nanoparticles (Rev-AuNPs) inhibited breast cancer progression via the suppression of NF-kB, AP-1, Akt, ERK, MMP-9, and COX-2 and the upregulation of HO-1 compared to RES alone in MCF-7 breast cancer cells. Furthermore, Song et al. [180] demonstrated that epidermal growth factor-modified docetaxel and RES co-encapsulated lipid-polymeric hybrid NPs (EGF DTX/RSV LPNs) significantly inhibited the viability and proliferation of HCC827 and NCIH2135 non-small cell lung cancer cells (NSCLCs) compared to DTX/RSV LPNs or free DTX/RSVDTX. Similarly, Wang et al. [181] reported that resveratrol-loaded solid lipid nanoparticles (RES-SLNs) induced enhanced apoptosis via the upregulated ratio of Bax/Bcl-2 and the suppression of cyclin D1 and c-Myc compared to RES alone in MDA-MB-231 cells. Karthikeyan et al. [182] reported that the anticancer effects of RES-loaded gelatin nanoparticles (RSV-GNPs) were improved in NCI-H460 cells compared to resveratrol alone via the upregulation of Bax, p53, p21, and caspase-3; the downregulation of Bcl-2 and NF-κB; and G0/G1 arrest. Similarly, Guo et al. [183] revealed that resveratrol–bovine serum albumin nanoparticles (RES-BSANP) induced apoptosis via the activation of apoptosis-inducing factor (AIF), cytochrome c, and Bax in SKOV3 ovarian cancer cells. Furthermore, Fan et al. [184] reported that resveratrol-loaded oxidized mesoporous carbon nanoparticles (oMCNs-RES) showed better cytotoxicity and promoted the drug loading efficiency and solubility of RES compared to resveratrol alone in MDA-MB-231 breast cancer cells. Overall, combining RES with NPs containing materials such as polymers, lipids, proteins, carbon nanotubes, and metals enhances the antitumor effects of RES by improving drug delivery and bioavailability, suggesting the potential for further clinical triald of RES NPs in cancer patients in the future. Overall, the potency of the nano-phytochemicals mentioned in this review has been verified, and thus, they are worthy of further development for clinical translation. Additionally, the molecular mechanism of action, pharmacokinetic profiles, toxicological profiles of nano-phytochemicals should be further explored, though nano-phytochemicals bring benefits in cancer therapy via efficient drug delivery, stability, EPR effect and low clearance by RES. 

## 5. Clinical Implications

Since nanotechnology can offer new solutions for the development of cancer diagnosis and therapeutics, several antitumor nanodrugs have been tested in preclinical and clinical trials over the past decade [186]. To date, seven anticancer nanodrugs, namely, Doxil, DaunoXome, Eligar, Abraxane, Marquibo, Onivyde, and Mycocet, have entered the pharmaceutical market, while Astragen (for acute promyelocytic leukemia), Lipoplatin (for pancreatic, head and neck, and breast cancers), Aurimmune (CYT-6091) (for head and neck cancer), and Paclical (for ovarian cancer) are still undergoing Phase II/III clinical trials [187,188].

Recently, several phytochemicals have been reported as promising anticancer candidates with significant efficacy in vitro and in vivo. However, many clinical trials have revealed that they lack efficacy in humans, mainly due to poor bioavailability, insolubility, instability, rapid clearance, and toxicity [95]. As a nano-phytochemical, albumin-bound paclitaxcel (PTX; Abraxane) derived from *Taxus brevifolia* was found be more effective in delivering PTX to tumors and reducing its toxicity in normal cells compared to PTX alone [103]. Furthermore, to overcome the poor bioavailability of curcumin in humans [120,131], combining it with piperine to block the metabolic pathway of curcumin [189,190], the use of a colloidal NP THERACURMIN [191] as a standardized curcuminoid mixture, the application of its corresponding lecithin formulation (Meriva) as a phospholipid complex [192], and structural analogs [193] have been suggested for efficient cancer therapy. As in recent nano-research that found that several anticancer phytochemical-encapsulated NPs were more effective compared to NP controls [139], Gota et al. [194] reported the potency of solid lipid curcumin NPs: 22.43 ng/mL curcumin was detected at a dose of 650 mg, while it was not detected in the plasma of osteosarcoma patients. Similarly, a clinical trial (NCT03140657) showed that the solubility of curcumin in nanomicelles was increased by 100 thousand times, and another clinical trial (NCT01403545) revealed that intravenous injection of liposomal curcumin was well tolerated and safe, except for the formation of transient red blood cell echinocytes at dosages ≥ 120 mg/m^2^ [195], implying the potential use of curcumin NPs for the clinical treatment of cancer in the future. However, though numerous studies with nanocurcumin formulations using liposomes, polymers, conjugates, cyclodextrins, micelles, dendrimers, and other nanoparticles have shown favorable outcomes in vitro and in vivo, clinical trials with nanocurcumin have not yet been conducted in cancer patients. Hence, to develop anticancer nano-phytochemical drugs, many preclinical and clinical trials should be conducted in a large number of cancer patients to gather information on the safety, toxicity, and efficacy of nano-phytochemical formulations alone or in combination with classical anticancer agents.

## 6. Conclusions and Perspectives

Over the past decade, the advanced progress in nanobiotechnology and nanomaterial science has contributed to imaging diagnosis and targeted cancer therapy by providing efficient drug delivery systems. A variety of organic nanoparticles (NPs) (such as polymeric conjugates, lipid-based carriers (liposomes and micelles), dendrimers, and ferritin) and inorganic NPs (such as carbon nanotubes, quantum dots, ceramic NPs, magnetic NPs, and gold/silver NPs) encapsulated with classical anticancer drugs or phytochemicals have been developed to promote enhanced permeability and retention (EPR) in cancer. However, smart nanoencapsulation designs and synthesis of potent anticancer phytochemicals are required to better understand the physicochemical properties of NPs, such as their shape, size, charge, surface chemistry, and toxicity, since many natural compounds, such as anacardic acid, betulinic acid, curcumin, EGCG, ferulic acid, gambogic acid, ginsenosides, kaempferol, lycopene, and resveratrol, have been found to be effective in culture cells and animals with low toxicity. However, many phytochemicals have not shown significant anticancer efficacy due to their poor solubility, stability, rapid clearance through RES phagocytosis, drug distribution, degradation, and resistance in humans [196]. In contrast, phytochemical-encapsulated NPs have been shown to enhance antitumor effects in several cancers via the inhibition of PI3K/AKT, NF-κB, ERK, and anti-apoptotic proteins (such as survivin, Bcl-2, and Bcl-_X_L) and the activation of PARP, caspases-8, - 9, and -3, BAX, and p53 due to enhanced bioavailability, solubility, stability, and EPR effects compared to free phytochemicals or NPs alone (Figure 3). Nonetheless, the difference between nano-phytochemicals and free phytochemicals or NPs alone for a specific molecular target should be further elucidated by evaluating the synergistic or additive effects between NPs and free phytochemicals. In addition, though the EPR effect works only in solid tumors via passive targeting, while active targeting through functionalization with ligands (including proteins, antibodies, and peptides) works in hematological malignancies, the complexity and the heterogeneity of tumor phenotypes should be considered for effective clinical translation [197]. Moreover, the median and mean delivery efficiencies are reported to only be 0.70% and 1.48% of the injected dose (ID) of NPs, respectively, which is considered a major drawback for clinical translation [198,199]. Hence, for the clinical translation of nano-phytochemicals, it is crucial to design EPR-potentiating combination therapy, evaluate the extent of the EPR effect, and utilize ultrasound, radiation, hyperthermia, and photodynamic therapy as a physical targeting strategy to achieve an efficient EPR effect in EPR-insensitive tumors before and during clinical trials [105]. Additionally, the reliability and reproducibility of NP–phytochemical encapsulations should be ensured, and their possible toxicity, suitable administration routes (oral, intravenous, transdermal), and pharmacodynamic and pharmacokinetic profiles should be determined in many preclinical and randomized clinical trials because some phytochemicals can have adverse side effects [200], though they are known to generally be less toxic compared to synthetic anticancer drugs [201]. Furthermore, further study is required to elucidate the detailed molecular mechanism of nanodrugs within the tumor environment for active drug delivery; prolong the circulation time; and explore the RES evasion, sensitive drug release, and targeted co-delivery of different compounds [202]. Finally, along with the above-mentioned challenges that must be overcome, an appropriate strategy for lowering manufacturing costs and scaling up the production of anticancer phytochemical candidates should be developed to ensure their success on the pharmaceutical market.

## Figures and Tables

**Figure 1 ijms-22-03571-f001:**
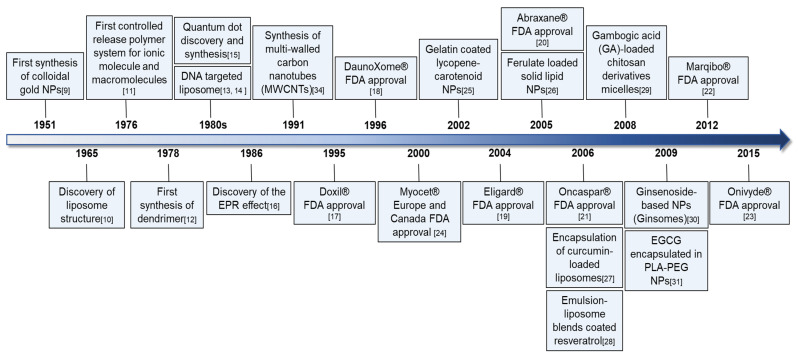
Research milestones of nanotechnology, anticancer nanodrugs, and nano-phytochemicals.

**Figure 2 ijms-22-03571-f002:**
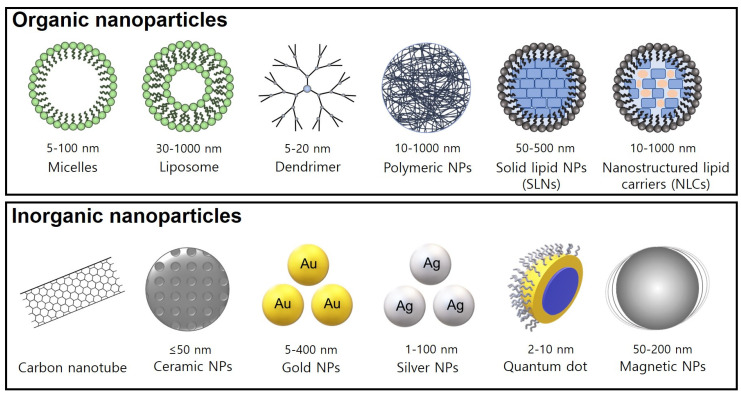
Morphology and size of nanostructured NPs. The physicochemical properties of these NPs include biocompatibility, biodegradability, and controlled/targeted drug release.

**Figure 3 ijms-22-03571-f003:**
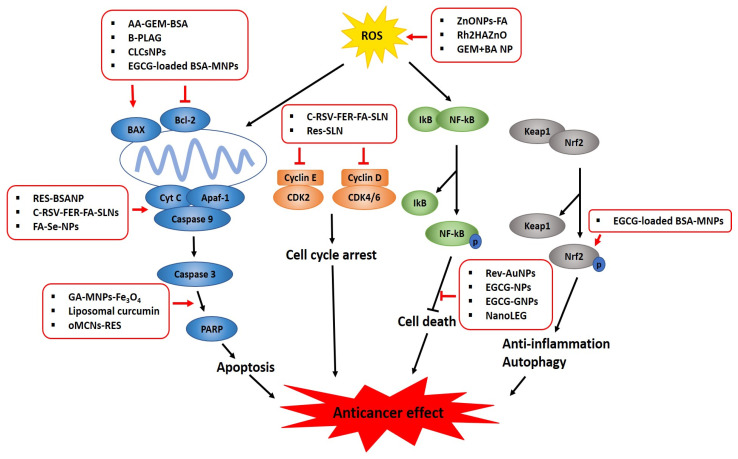
Graphical abstract on the molecular mechanism of nano-phytochemicals in cancers.

**Table 1 ijms-22-03571-t001:** Efficacy and molecular mechanism of nano-phytochemicals in several cancers.

Botanical Name	Natural Compound	Structure	Type of Nanoparticles	Efficacy	Mechanism	Concentration	Cancer	Cell Line	References
*Anacardium* *occidentale*	Anacardic acid	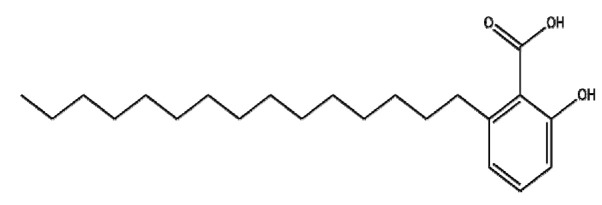	Protein (Docetaxel-loaded AA-GEM-BSA)	Apoptosis↑ Tumor size↓	HSP90, Hsp70, GRP78, Hsp70, CDK-4, MMP-9, Bcl-2, and Mcl-1↓	AA-GEM-BSA NP: 10, 15, 20 ug/mL; BSA NP: 10, 15, 20 ug/mL; In vivo: 0.5 mg/mL	Breast cancer in vivo	MCF-7, MDA-MB-231 DMBA induced breast cancer model	[125]
*Betula pubescens*	Betulinic acid	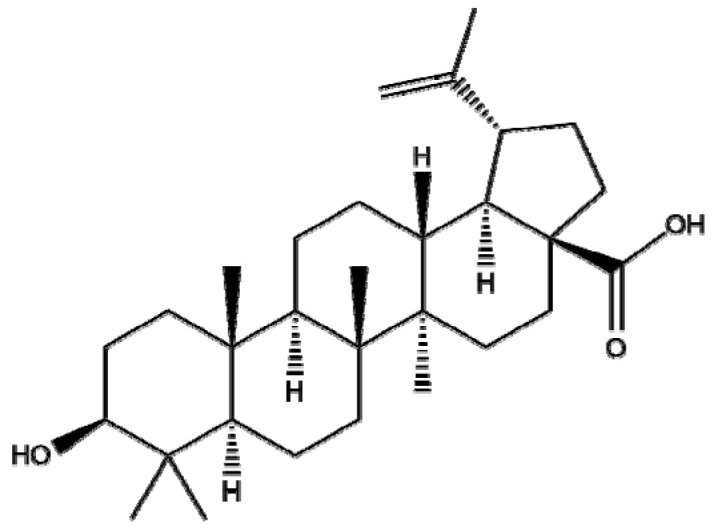	Polymer (PLGA-mPEG)	Apoptosis↑ ROS↑ tumor volume↓	ROS↑	GEM NP: 17.59 ± 2.91 ng/mL GEM+BA NP: 7.62 ± 0.84 ng/mL 2~8.5 mg/kg	Pancreatic cancer in vivo	Panc1 Ehrlich tumor model	[128]
			Polymer (B-PLAG)	Apoptosis↑	i-NOS↓ Bcl-2↓, Bcl-xl↓, BAD↑ → c-caspase3, c-caspase9↑	Betulinic acid: 100 mg/kg B-PLAG: 100 mg/kg	Liver cancer	in vivo	[130]
			Lipid (BA-loaded magnetoliposomes)	Apoptosis↑ Migration↓		25 μM	Breast cancer	MDA-MB-231, MCF7	[129]
*Curcuma longa*	Curcumin	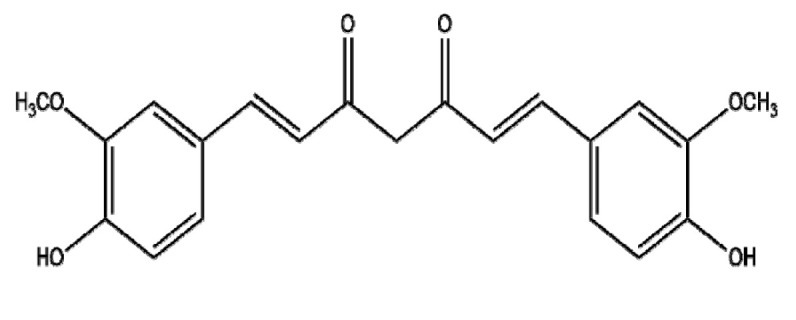	Polymer (CLCs)	Proliferation↓ Apoptosis↑	Bax ↑, Bcl-2 ↓	CLCsNPs: Caski→154.59 ug/mL, C33A→144.77 ug/mL, Hela→166.49 ug/mL, SiHa→188.55 ug/mL	Cervical cancer	Caski, C33A, HeLA, SiHa	[136]
			Polymer (chitosan/PEG-CUR)	Apoptosis↑ Invasion↓ Migration↓	Bax ↑, Bcl-2 ↓→ c-caspase-3 ↑ → c-PARP ↑	Curcumin: 10 uM chitosan/PEG-CUR: 10 uM	Pancreatic cancer	PANC-1, Mia Paca-2	[135]
			Lipid (Liposomal-CUR)	Proliferation↓ Apoptosis↑ Angiogenesis↓	CD31↓, VEGF↓, IL-8↓, PARP↑	10 μmol/L In vivo: 40 mg/kg	Colorectal cancer	Colo205 LoVo	[133]
			Lipid (EPC-CUR)	Apoptosis↑		molar ratio of curcumin/EPC 1:14	Colorectal cancer	HCT116, HCT15	[134]
			Lipid curcumin-resveratrol-gelucire 50/13-HPβCD (CRG-CD)	Apoptosis↑ Proliferation↓		CUR-RES-gelucire 50/13-HPβCD : 9.9 μM CUR-RES-gelucire 50/13: 6.9 μM	Colorectal cancer	HCT116	[137]
*Camellia sinensis*	EGCG	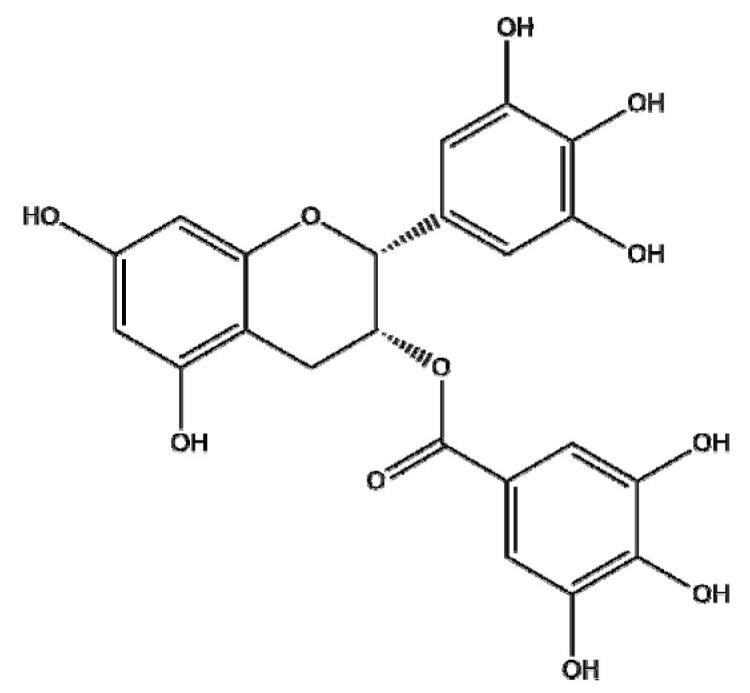	Polymeric (EGCG-Loaded FU/HA/PEG-Gelatin NPs)	Apoptosis↑ Anti-tumor efficacy↑		FU/HA/PEG-gelatin/EGCG:CU = 0.600:0.600:3.750:1.000:0.025	Prostate cancer	Luc PC3	[139]
			Polymer (EGCG-PLGA-NP)	Apoptosis↑ Proliferation↓	NF-κB↓→ Bcl-2, Bcl-xL, COX-2, TNF-α, cylcinD1, c-Myc, TWIST1, MMP-2↓	EGCG: 12.5, 25 μM EGCG-NPs: 12.5, 25 μM	Lung cancer	A549, H1299	[140]
			Lipid (EB-SLN)	Apoptosis↑ Migration↓		EGCG: 65.4 ± 4.9 μg/mL EGCG-SLN: 6.9 ± 1.1 μg/mL EB-SLN: 3.2 ± 1.7 μg/mL	Breast cancer	MDA-MB-231, B16F10 in vivo (C57/BL6 mice)	[142]
			Lipid (NLC-RGD)	Apoptosis↑		EGCG-loaded NLC-RGD: 45 μg/mL	Breast cancer	MDA-MB-231	[143]
			Protein (EGCG-loaded BSA-MNPs)	Apoptosis↑ ROS↑	Nrf2↑→Keap1↓→HO-1↑→ Bcl-2↓, Bax↑, Bak↑, Bim↑, puma↑	EGCG-loaded BSA-MNPs: 8 μM	Lung cancer	A549	[141]
			Metal (GNPs)	Apoptosis↑	Bcl-2↓, Bcl-xL↓, Bax, c-caspase7↑, c-caspase3↑ NF-kb/p65↓ (Nuclear)	50 μg/mL	Breast cancer Prostate cancer	MCF10-A, MDA-MB-231, RWPE1, PC3	[144]
*Ferula communis*	Ferulic acid	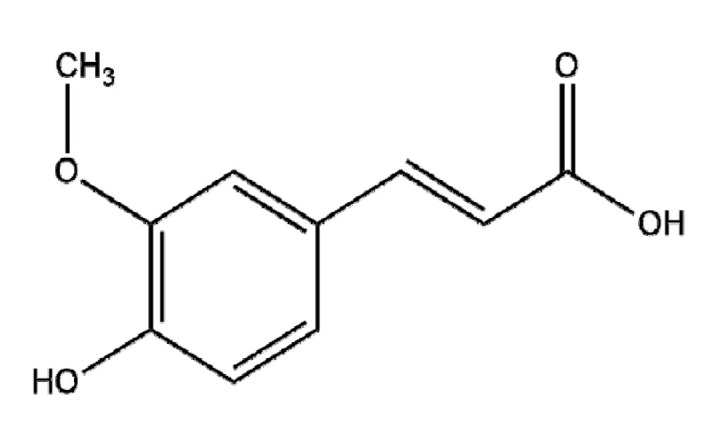	Polymer (chitosan-SLNs)	Apoptosis↑ Tumor growth↓	PCNA, Ki67↓, p-ERK1/2↑→p21, p-Rb ↑	In vitro: 40 and 25 μM of FA and ASPIn vivo: 75 and 25 mg/kg of FA and ASP	Pancreatic cancer	PANC-1, MIA-PaCa-2	[150]
			Metal (FA-Se)	Apoptosis↑ ROS↑	MMP↓ Caspase 3, Caspase 9 activation ↑	FA-Se-NPs: 5, 10, 20 μg/ml	Liver cancer	HepG2	[148]
			Metal (ZnO)	Cell cycle arrest↑ Apoptosis↑ Nodular formation↓	ROS↑, MMP↓, DNA damage↑→ Bcl-2↓, Bcl-xL↓, Bax↑, Bad↑, c-caspase-3↑, c-PARP↑,		Liver cancer	HepG2, Huh-7	[149]
*Garcinia hanburyi*	Gambogic acid	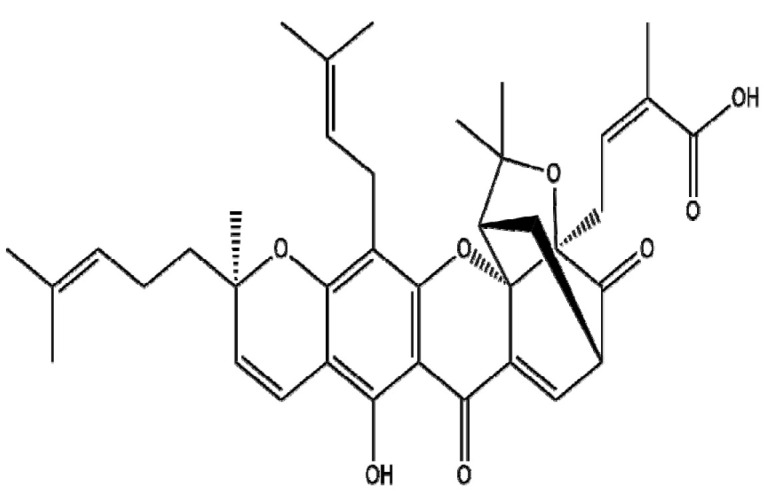	Polymer (Met-PEA-PEG)	ROS↑ Proliferation↓ Apoptosis↑		0.1~1.0 μg/mL	Prostate cancer Cervical cancer embryo	PC3, Hela, NIH 3T3	[154]
			Polymer (PEI-PLGA)	Proliferation↓ Apoptosis↑	pTRAIL/GA-HA/PPNPs → caspase3↑, caspase8↑, survivin↓, Bcl-2↓	0.0125~1 μM	Breast cancer	MCF-7, MDA-MB-231, 4T1	[155]
			Polymer (Arg-PEUUs)	Migration ↓ Invasion ↓	FA/Arg/GA/PEUU → MMP-2, MMP-9↓	GA: 0.6 µg/mL	Cervical cancer Lung cancer Colorectal cancer	Hela, A549 HCT116	[156]
			Metal (Fe3O4)	Proliferation↓ Apoptosis↑	PI3K↓→ Akt↓→Bad↓→caspase9,3↑	0.25~0.75 µg/mL	Colorectal cancer	LoVo	[157]
*Panax ginseng*	Ginsenoside Rg3	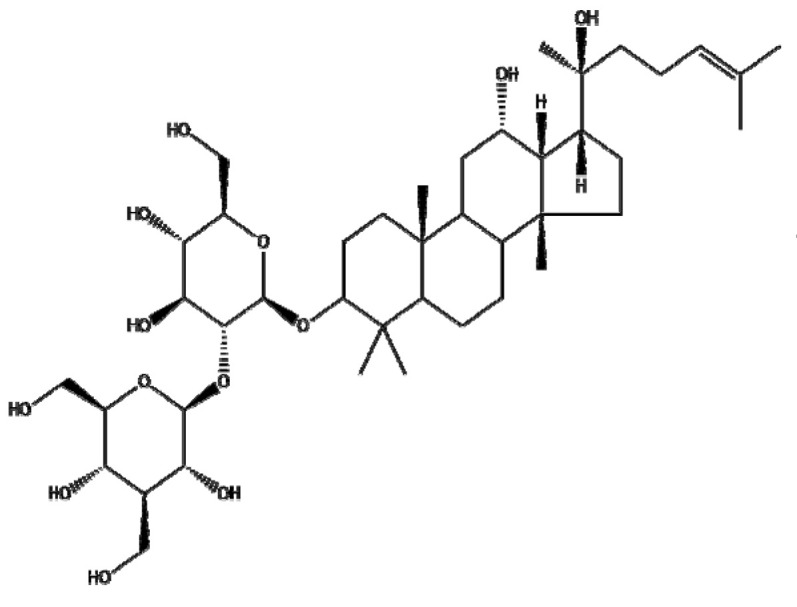	Lipid (mPEG-b-P(Glu-co-Phe))	Apoptosis ↑ Proliferation↓	PCNA↓, caspase3↑	Rg3-NPs: 50 μg/mL ^−1^	Colorectal cancer	SW480, SW620, CL40, CCD-18Co	[162]
			Metal (Fe_3_O_4_-NpRg3)	Survival↑ Proliferation↓ Invasion↓ Immune response↑		Fe_3_O_4_ : 70 mg kg^−1^ Rg3 : 70 mg kg^−1^ Fe_3_O4-Rg_3_ : 70 mg kg^−1^	in vivo	in vivo	[161]
	Ginsenoside Rg5	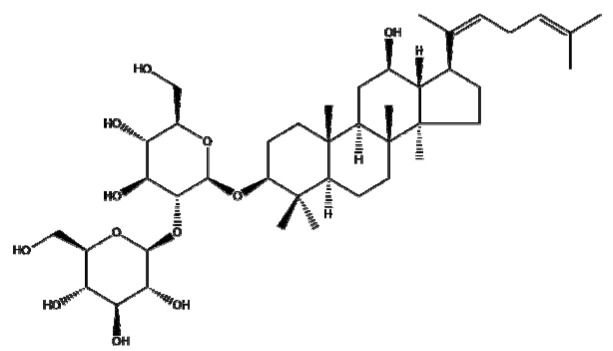	Protein (FA-Rg5-BSA NP)	Apoptosis ↑ Proliferation↓ Tumor growth ↓ Cellular uptake↑	Rg5-BSA NP(pH7.4) < FA-Rg5-BSA NP(pH7.4)	FA-Rg5-BSA: 50 μM, 0.5 mg/kg Rg5-BSA: 50 μM, 0.5 mg/kg	Breast cancer Mouse fibroblast cell	MCF-7 L929	[163]
	Ginsenoside Rh2	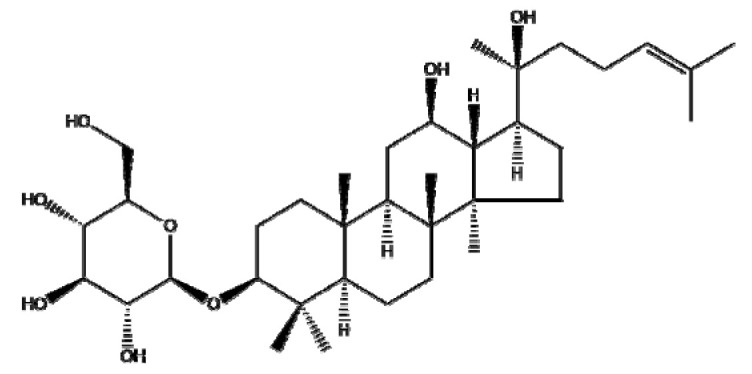	Carbon (Rh2HAZnO)	ROS↑ Apoptosis ↑	MAPK, p38↑, p53↑→caspase7↑, caspase9↑, BAX↑, ROS↑, PARP↑	Rh2HAZnO: 20 µg/mL	Lung cancer Colorectal cancer Breast cancer	A549, HT29, MCF7	[44]
*Crocus sativus*	Kaempferol	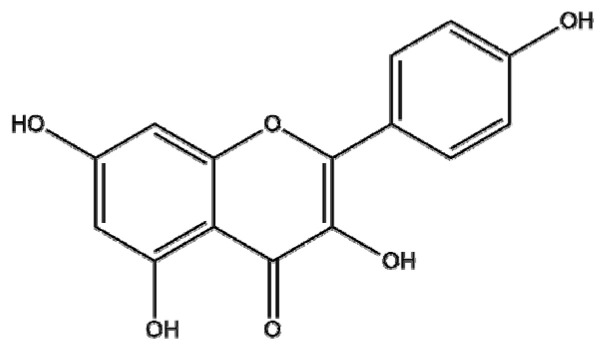	Polymer (kaempferol-PEO-PPO-PEO, kaempferol-PLAG)	Proliferation↓	Efficacy: Kaempferol < K-PEO-PPO-PEO,K- PLAG	PEO-PPO-PEO: 25 μM PLGA: 25 μM PLAG-PEI: 25 μM Chitosan: 25 μM PAMAM: 25 μM Kaempferol: 25 μM	Ovarian surface epithelial cell Ovarian cancer	IOSE397, A2780/CP70, OVCAR-3	[166]
			Lipid (KPF-MNE)	Proliferation↓ Cellular uptake↑ Apoptosis↑		KPF: 1 μM KPF-MNE: 1 μM	Glioma cell	C6 rat ex vivo in vivo	[169]
			Metal (K-AuNCs)	Proliferation↓ Cytotoxicity↑		K-Au: 12.5 μg/mL	Lung cancer	HK-2 A549	[168]
			Gelatin (GNP-KA)	Proliferation↓ Angiogenesis↓	MMP-2, MMP-9, VEGF↓	KA: 7.4 μg/mL GNP-KA: 150 μg/mL	Human umbilical vascular cells	HUVEC in vivo	[167]
*Solanum lycopersicum*	Lycopene	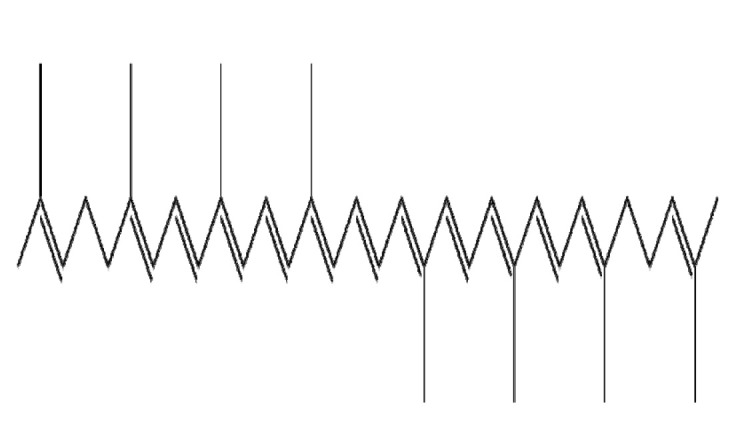	Polymeric (LYC-WPI-NPs)	Apoptosis↑ Bioavailability↑ ROS↑		LYC-WPI-NPs: 15.0 mg/kg/animal	in vivo	MCF-7 in vivo	[173]
			Lipid (poly-ɛ-caprolactone lipid-core)	ROS↑	ROS↑→ NF-κB↓	NanoLEG: 200 μg/mL	Breast cancer	MCF-7 HMC-3	[172]
			Metal (rGO-AgNPs) + TSA	ROS↑ Apoptosis↑ DNA damage↑	MDA↑ GSH↓ MMP↓ p53↑→Bax↑, Bak↑→Bcl-2↓ Rad51↑→ γH2AX↑	rGO-Ag: 0.30 µM 8TSA: 0.20 µM	Ovarian cancer	SKOV3	[185]
			Metal (LP–AN)	Apoptosis↑	BcL-2↓ Bax↑→ caspase 8, 3, 9, PARP-1↑ AKT↓→β-catenin↓→ NF-κB↓→MMP-2, MMP-9↓	AN: 0.16 ppm LP: 0.4 μM	Colorectal cancer Lung fibroblast	HT-29 MRC-5	[174]
*Vitis vinifera*	Resveratrol	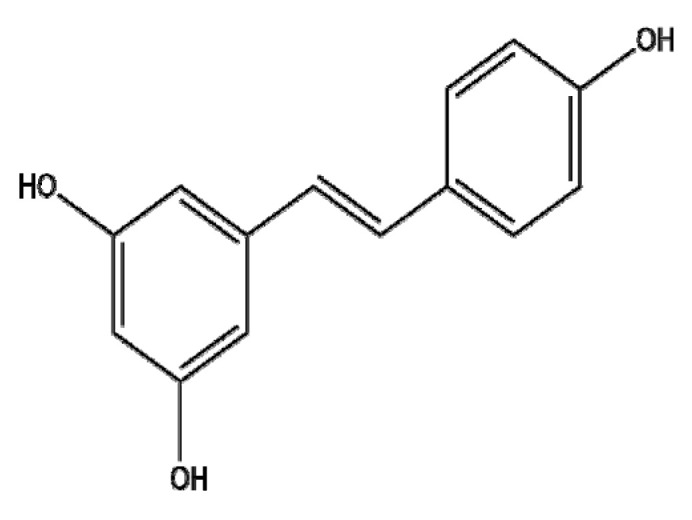	Polymer (Chitosan-coated RSV-FER-FA-SLNs)	Apoptosis↑Cell cycle arrest↑	CyclinD1, cdk2, cdk4, cdk6, cyclinE, cyclinB↓ cytochrome C↑→ c-caspase9↑→ c-caspase3↑	C-RSV-FER-FA-SLN: 10 μg/mL RSV-FER-FA-SLN: 25 μg/mL	Colorectal Cancer	HT-29	[178]
			Lipid (EGF DTX/ RSV LPNs)	Apoptosis↑ Proliferation ↓		In vivo: EGF DTX/RSV LPN: 50 mg/kg In vitro: 0~100 μg/mL	Lung cancer	HCC827, NCIH2135 in vivo	[180]
			Lipid (Res-SLN)	Proliferation ↓ Migration ↓ Invasion ↓ Apoptosis ↑	c-Myc↓→ Bax/Bcl-2 ↑ → cyclinD1↓	Resveratrol: 40 μM Res-SLN: 40 μM	Breast cancer	MDA-MB-231	[181]
			Protein (RSV-GNP)	ROS↑ Cell cycle arrest ↑ Apoptosis ↑	p53 ↑→ Bax/Bcl-2, c-cas-9, -3, p21↑		Lung cancer	NCI-H460	[182]
			Protein (RES-BSANP)	Apoptosis ↑	AIF↑, cytochrome c ↑ → Bax ↓	RES-BSANP: 50 μM	Ovarian cancer	SKOV3	[183]
			Metal (Rev-AuNPs)	Invasion↓, migration↓,	NF-κB↓→ AP-1↓→ MMP-9↓, COX-2↓ Akt↓, ERK↓→ MMP-9↓, COX-2↓ HO-1↑→ MMP-9↓	Rev: 10 μM Rev-AuNPs: 10 μM	Breast cancer	MCF-7	[179]
			carbon (oMCNs-RES)	Proliferation↓ Apoptosis↑	c-caspase3↑, c-PARP↑	Resveratrol: 100 μM oMCNs-RES: 200 μM	Breast Cancer	MDA-MB-231	[184]

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
