# Peer review of "Recent Advances in Nanotechnology with Nano-Phytochemicals: Molecular Mechanisms and Clinical Implications in Cancer Progression"

_ijms, 2021, doi:10.3390/ijms22073571_

Round 1

Reviewer 1 Report

I read with great interest the Review manuscript entitled “Recent advances in nanotechnology with nanophytochemicals, their molecular mechanism and clinical implications in cancer progression”.

The Authors discussed the progress of nanotechnology, research milestones and molecular mechanism of phytochemicals encapsulated nanoparticles, clinical implications and finally suggested many challenges to overcome and future research perspectives. The manuscript is well written and detailed. It includes helpful images. Only minor ortographical and grammatical errors have been found throughout the manuscript.

Author Response

Thanks. Our MS was edited by MDPI team. Attached was English edited certificate.

Reviewer 2 Report

In this manuscript the authors describe the molecular mechanism of nano-phytochemicals, their clinical implications and future research perspectives.  The work seems well executed. However, the work should be publishable subsequently to some consideration of the following major comments.

Majors concerns

The authors provide a long description of different NPs associated with phytochemical. However, NPs may show undesirable off-target effects. The toxicological profiles and mechanisms of cytotoxicity of NPs and nanophytochemicals, should be better investigated in order to suggest the potential application of NPs in medicine.  The authors provide a summary of each source related to the topic, which results in an annotated bibliography in absence of a critical evaluation of data.

Some critical points as “EPR effect”, the “implication of an “EPR-insensitive tumor phenotypes, the active targeting and physical targeting methods should be more deeply discussed. The authors should also critically discuss some important points such as NPs stability and clearance.

Moreover, the authors should investigate better how nano-phytochemicals bring benefits in cancer therapy. To this end, the molecular mechanism of action, pharmacokinetic profiles, toxicological profiles of nano-phytochemicals should be critically discussed by the authors in section 4 in order to help the readers to better understand perspective of nano-phytochemicals in the pharmaceutical market.

Author Response

In this manuscript the authors describe the molecular mechanism of nano-phytochemicals, their clinical implications and future research perspectives.  The work seems well executed. However, the work should be publishable subsequently to some consideration of the following major comments.

Majors concerns

The authors provide a long description of different NPs associated with phytochemical. However, NPs may show undesirable off-target effects. The toxicological profiles and mechanisms of cytotoxicity of NPs and nanophytochemicals, should be better investigated in order to suggest the potential application of NPs in medicine.  The authors provide a summary of each source related to the topic, which results in an annotated bibliography in absence of a critical evaluation of data.

(Response) Thanks for critical comments. We added the necessity of toxicology and PK profile of NPs and nanophytochemicals for efficient clinical application.

Some critical points as “EPR effect”, the “implication of an “EPR-insensitive tumor phenotypes, the active targeting and physical targeting methods should be more deeply discussed. The authors should also critically discuss some important points such as NPs stability and clearance.

(Response) Thanks. More discussed in section 3.9.

Moreover, the authors should investigate better how nano-phytochemicals bring benefits in cancer therapy. To this end, the molecular mechanism of action, pharmacokinetic profiles, toxicological profiles of nano-phytochemicals should be critically discussed by the authors in section 4 in order to help the readers to better understand perspective of nano-phytochemicals in the pharmaceutical market.

(Response) Thanks. More discussed in the end of section 4.

Round 2

Reviewer 2 Report

This second version of this manuscript is improved. However, the authors should comment betterv the “EPR effect” and  the “implication of an “EPR-insensitive tumor phenotypes,

Author Response

Thanks for your kind comments to improve the quality of our MS. Based on your comments, more information was added on EPR effect and EPR insensitive phenotypes in Section 3-9.